# Hypoperfusion states could increase the risk of non-arteritic anterior ischemic optic neuropathy

Jasmin Gabbay[1☯], Eyal Walter[2☯], Tomer Kerman[1,3], Nir Amitai[1,3], Ohad Gabay[4], Itai Hazan[1,3], Ran Abuhasira[3], Erez Tsumi[2]*

1 Joyce & Irving Goldman Medical School, Ben-Gurion University of the Negev, Be'er-Sheva, Israel, 2 Department of Ophthalmology, Soroka University Medical Center, Ben-Gurion University of the Negev, Be'er-Sheva, Israel, 3 Clinical Research Center, Soroka University Medical Center and Faculty of Health Sciences, Ben-Gurion University of the Negev, Be'er-Sheva, Israel, 4 Department of Intensive Care, Soroka University Medical Center, Ben-Gurion University of the Negev, Be'er-Sheva, Israel

☯ These authors contributed equally to this work.
* ertsumi@post.bgu.ac.il

## Abstract

### Background

Non-arteritic anterior ischemic optic neuropathy (NAION) is a leading cause of acute optic nerve damage. While cardiovascular risk factors such as hypertension, diabetes mellitus, and obstructive sleep apnea are well-established, the association between NAION and states of hypoperfusion is underexplored. This study investigated this potential association.

### Methods

This retrospective case-control study analyzed all electronic medical records of Clalit Health Services' patients from 2001 to 2022. Patients diagnosed with NAION were matched in a 1:4 ratio by year of birth and sex, using propensity score analysis to adjust for various comorbidities. Events of hypoperfusion occurring in the month prior to the diagnosis of NAION were categorized into two physiological mechanisms: a decrease in SVR and a decrease in cardiac output due to cardiac dysfunction or diminished preload (attributed to hypovolemia). Conditional logistic regression was used to explore differences between the groups.

### Results

A total of 1,374 patients diagnosed with NAION and 5,496 matched controls were included in the study. We found a nearly 6.5-fold increase in the likelihood of NAION in association with events of hypoperfusion that occurred in the month period preceding the diagnosis of NAION (odds ratio [OR] 6.48; 95% confidence interval [CI]: 5.05–8.32). In particular, the group of patients with cardiac dysfunction (OR 6.47; 95% CI: 4.63–9.04) and the group with hypovolemia (OR 6.1; 95% CI: 4.08–9.13) emerged as having the most substantial risk factors. The group with decreased Systemic Vascular Resistance (SVR) (OR 4.64; 95% CI:

**Data Availability Statement:** Data are available on request due to potentially sensitive patient information. For requests and more information,

please contact Lital Abuzaglo (liabuzaglo1@clalit.org.il).

**Funding:** The author(s) received no specific funding for this work.

**Competing interests:** The authors have declared that no competing interests exist.

2.84–7.59) was also strongly related with NAION. Cerebrovascular accident emerged as an independent significant risk factor for NAION (OR 16.1; 95% CI: 10.8–24).

## Conclusion

Hypoperfusion states are significant, independent risk factors for NAION.

## Introduction

Non-arteritic anterior ischemic optic neuropathy (NAION) is the most common cause of acute optic nerve damage in patients over the age of 50, with incidence peaking around 60 years of age. The annual incidence varies but ranges from 2.3 to 10.2 cases per 100,000 patients, with a slight male predominance [1, 2]. NAION typically manifests as painless, sudden loss of vision [1, 2]. The exact cause of NAION is uncertain, but it is thought to be the result of reduced perfusion of the short posterior ciliary arteries, leading to ischemia and swelling of the optic nerve head. Being wrapped by the rather inelastic sclera, this swelling leads to the development of compartment syndrome, further reducing perfusion and resulting in apoptosis of the ganglion cells that comprise the optic nerve and their axons [1].

Chronic cardiovascular comorbidities such as hypertension, diabetes mellitus, hyperlipidemia, anemia and obstructive sleep apnea are well established risk factors for NAION [1–8]. An important risk factor for NAION is an anatomically crowded optic nerve head with small optic cups, also known as "disc at risk", which probably indicates the optic nerve head has less room for expansion [1, 2, 4]. Nocturnal hypotension may also contribute to NAION, which may explain the symptoms observed upon awakening and could possibly imply an association with hypoperfusion [1, 7].

The association of NAION and cardiovascular risk factors has been well-explored. In his work, Hayreh investigated these risk factors, classifying them as "predisposing factors", while "precipitating factors" were identified as primarily related to arterial hypotension [8]. However, the relationship between NAION and hypoperfusion states has only been examined in small scale studies. This large scale, nationwide study was conducted to investigate the association between different states of hypoperfusion such as decreased cardiac output (due to hypovolemia or cardiac dysfunction) and systemic vascular resistance (SVR) and NAION.

## Methods

### Study population

This retrospective case-control study utilized electronic medical records from Clalit Health Services (CHS) in Israel, examining data from January 1, 2001 to December 31, 2022. Only individuals who had been members of CHS for at least one year were included, ensuring a reliable and consistent study population. As the largest health maintenance organization in Israel, CHS insures and provides medical services to approximately 4.8 million people; 51% of the Israeli population [9].

The case group (group N) included patients diagnosed with NAION, identified by the International Classification of Diseases (ICD-9) code 377.41. To ensure accuracy in case selection and to exclude conditions that could mimic NAION, participants with a record of giant cell arteritis (ICD-9 code 446.5) or optic neuritis (ICD-9 code 377.30) diagnosis were excluded from the case group.

For each individual in group N, four controls were matched by birth year and sex, comprising group C. The matching was further refined using propensity score analysis to adjust for specific comorbidities, including liver disease, diabetes, renal disease, malignancy, and chronic pulmonary disease, based on ICD-9 codes, as detailed in S1 Table.

## Variable definitions

Demographic data and pre-existing comorbidities for each patient were recorded using the ICD-9 codes listed in S1 Table.

Hypoperfusion-related events within one-month preceding the diagnosis of NAION for patients in Group N were assessed. For individuals in Group C, the reference date for this evaluation was aligned with the NAION diagnosis date of the matched counterparts in Group N. Hypoperfusion events were categorized into two physiological mechanisms: a decrease in SVR and a decrease in cardiac output due to cardiac dysfunction or diminished preload (attributed to hypovolemia). The hypovolemia group included major surgeries that are highly likely to precipitate a hypotensive episode. The conditions falling under each of these categories are described in Tables 3–5.

Additionally, conditions related to hypoperfusion that did not fit into a single category, such as pneumothorax, syncope, aortic dissection, and shock were included in separate group and defined as Other. Cerebrovascular accidents (CVA) distinct from NAION and that preceded its onset were also evaluated. All events were labeled using their respective ICD-9 codes or relevant surgical codes, as detailed in S1 Table.

## Statistical analysis

Continuous variables were summarized using means, medians, standard deviations, minimums and maximums, while categorical variables were represented by frequencies and percentages. A conditional logistic regression model was used to identify differences between groups N and C. This regression model was selected for its effectiveness in handling the correlations intrinsic to the matched case-control study design. All statistical tests were two-tailed, and a p-value of less than 0.05 was considered statistically significant. The analyses were performed using R software, version 3.6.1.

## Ethics approval

This case-control study was approved by the CHS Research Ethics Committee and the Soroka Medical Center Institutional Review Board on May 1, 2023 (approval number: SOR-0198-23). The study was conducted in accordance with the principles of the Declaration of Helsinki. Patient informed consent was not required for this retrospective study that used de-identified data.

## Data collection

Data were extracted via the CHS Data Sharing Platform, powered by MDClone (https://www.mdclone.com). This platform uses advanced algorithms to de-identify data from electronic medical records, ensuring both data integrity and patient privacy. The authors accessed the data on December 12, 2023. They did not have access to information that could identify individual participants during or after data collection.

**Table 1. Baseline characteristics after 1:4 propensity score matching, by the study group.**

| Characteristic | NAION n = 1,374 | Controls n = 5,496 | Standardized mean difference |
|---|---|---|---|
| Age, years | | | 0 |
| Mean ± SD (N) | 67 ± 13 | 67 ± 13 | |
| Median (IQR) | 68 (59, 76) | 68 (59,76) | |
| Range | 4–100 | 4–100 | |
| Male sex n (%) | 783 (57) | 3,132 (57) | 0 |
| Systemic comorbidities, n (%) | | | |
| Liver disease | 126 (9.2) | 480 (8.7) | -0.0016 |
| Diabetes mellitus | 594 (43) | 2,366 (43) | 0.0003 |
| Renal disease | 204 (15) | 808 (15) | -0.0002 |
| Malignancy | 157 (11) | 616 (11) | 0.0054 |
| Chronic pulmonary disease | 357 (26) | 1,472 (26) | -0.0026 |

SD, standard deviation; IQR, interquartile range

## Results

A total of 1,374 individuals diagnosed with NAION (group N) were matched to 5,496 controls (group C). The patients' characteristics and systemic comorbidities of the matched groups are shown in Table 1. The average age of patients in both groups was 67 ± 13 years, with male predominance (57%). After propensity score matching, the standardized mean differences were less than 0.1 for all variables assessed. For a detailed examination of additional underlying comorbidities and demographic data not included in the matching process, refer to S2 Table.

The primary analysis revealed a significantly higher overall occurrence rate of hypoperfusion within the month leading to the diagnosis of NAION in group N compared to group C (odds ratio [OR] 6.48, 95% confidence interval [CI] 5.05–8.32; Table 2). This elevated risk extended across several physiological mechanisms, including decreased SVR (OR 4.64, 95% CI 2.84–7.59), cardiac dysfunction (OR 6.47, 95% CI 4.63–9.04), and hypovolemia (OR 6.1, 95% CI 4.08–9.13). In addition to these broad categories, group N also showed increased occurrence rates for specific conditions such as CVA (OR 16.1, 95% CI 10.8–24) and syncope (OR 5.78, 95% CI 2.47–13.5).

Further analyses were conducted to investigate the components of hypoperfusion-related conditions observed in group N. Each condition was examined within its respective

**Table 2. Assessment of hypoperfusion-related conditions one month prior to NAION onset and in matched controls.**

| Characteristic | NAION n = 1,374 | Controls n = 5,496 | Odds Ratio | 95% CI |
|---|---|---|---|---|
| Hypoperfusion condition, n (%) | 175 (13) | 131 (2.4) | 6.48 | 5.05–8.32 |
| Decrease in systemic vascular resistance | 36 (2.6) | 34 (0.6) | 4.64 | 2.84–7.59 |
| Cardiac dysfunction | 93 (6.8) | 63 (1.1) | 6.47 | 4.63–9.04 |
| Hypovolemia | 61 (4.4) | 42 (0.8) | 6.1 | 4.08–9.13 |
| Others | | | | |
| Pneumothorax | 2 (0.1) | 2 (<0.1) | 4 | 0.56–28.4 |
| Syncope | 13 (0.9) | 9 (0.2) | 5.78 | 2.47–13.5 |
| Aortic dissection | 4 (0.3) | 2 (<0.1) | 4 | 0.56–28.4 |
| Shock | 1 (<0.1) | 2 (<0.1) | 2 | 0.18–22.1 |
| Cerebrovascular accident, n (%) | 123 (9) | 34 (0.6) | 16.1 | 10.8–24 |

**Table 3. Assessment of conditions associated with decrease in systemic vascular resistance one month prior to NAION onset and in matched controls.**

| Characteristic | NAION n = 1,374 | Control, n = 5,496 | Odds Ratio | 95% CI |
|---|---|---|---|---|
| Decrease in SVR, n (%) | 36 (2.6) | 34 (0.6) | 4.64 | 2.84–7.59 |
| Sepsis | 3 (0.2) | 3 (<0.1) | 4 | 0.81–19.8 |
| Septic shock | - | 2 (<0.1) | - | - |
| Pyelonephritis | - | - | - | - |
| Peritonitis | 1 (<0.1) | 1 (<0.1) | 4 | 0.25–64 |
| Septic arthritis | - | - | - | - |
| Necrotizing fasciitis | - | - | - | - |
| Pneumonia | 27 (2) | 26 (0.5) | 4.45 | 2.54–7.79 |
| Cholecystitis | 5 (0.4) | 3 (<0.1) | 6.67 | 1.59–27.9 |
| Cholangitis | - | 1 (<0.1) | - | - |
| Anaphylactic shock | - | - | - | - |
| Vasopressor support | 3 (0.2) | 4 (<0.1) | 3 | 0.67–13.4 |

physiological category, during the same one-month period. Table 3 focuses on conditions associated with decreased SVR, highlighting a significant increase in the occurrence of pneumonia (OR 4.45, 95% CI 2.54–7.79) and cholecystitis (OR 6.67, 95% CI 1.59–27.9) in group N relative to group C, in the month before the diagnosis of NAION.

Table 4 delineates conditions contributing to cardiac dysfunction. Myocardial infarction (OR 6.27, 95% CI 4.43–8.86) and pulmonary embolism (OR 9.33, 95% CI 2.41–36.1) were notably higher in group N compared to group C in the month leading to the diagnosis of NAION.

Lastly, Table 5 details the analysis of conditions related to hypovolemia. This table reveals that gastrointestinal bleeding (OR 9.23, 95% CI 4.03–21.1), trauma (OR 2.75, 95% CI 1.28–5.93), hemodialysis (OR 12, 95% CI 1.25–115), and major surgery (OR 8.86, 95% CI 4.71–16.6) were all significantly more prevalent in Group N.

## Discussion

This research presents the results of a large-scale national study conducted to investigate the role of hypoperfusion states associated with NAION. We found a greater than sixfold increase in the occurrence of hypoperfusion episodes in the month preceding the diagnosis of NAION.

Cardiac dysfunction as a group had the highest correlation with NAION, and patients in the NAION group were found to be 6.47 times more likely to have had a diagnosis related to cardiac dysfunction in the month preceding the diagnosis when compared to the control group. Pulmonary embolism as an entity demonstrated the highest association (9.33-fold increase), followed by myocardial infarction (6.27-fold increase). To the best of our

**Table 4. Assessment of conditions associated with cardiac dysfunction one month prior to NAION onset and in matched controls.**

| Characteristic | NAION n = 1,374 | Controls n = 5,496 | Odds ratio | 95% CI |
|---|---|---|---|---|
| Cardiac dysfunction, n (%) | 93 (6.8) | 63 (1.1) | 6.47 | 4.63–9.04 |
| Cardiac arrest | - | 1 (<0.1) | - | - |
| Myocardial infarction | 85 (6.2) | 59 (1.1) | 6.27 | 4.43–8.86 |
| Acute pulmonary edema | 2 (0.1) | 1 (<0.1) | 8 | 0.73–88.2 |
| Pulmonary embolism | 7 (0.5) | 3 (<0.1) | 9.33 | 2.41–36.1 |

**Table 5. Assessment of conditions associated with hypovolemia one month prior to NAION onset and in matched controls.**

| Characteristic | NAION n = 1,374 | Controls n = 5,496 | OR | 95% CI |
|---|---|---|---|---|
| Hypovolemia, n (%) | 61 (4.4) | 42 (0.8) | 6.1 | 4.08–9.13 |
| Gastrointestinal bleeding | 19 (1.4) | 9 (0.2) | 9.23 | 4.03–21.1 |
| Trauma | 11 (0.8) | 16 (0.3) | 2.75 | 1.28–5.93 |
| Hypotension | 0 | 6 (0.1) | - | - |
| Hemodialysis | 3 (0.2) | 1 (<0.1) | 12 | 1.25–115 |
| Major surgery | 31 (2.3) | 14 (0.3) | 8.86 | 4.71–16.6 |
| Coronary artery bypass graft | 11 (0.8) | 5 (<0.1) | 8.8 | 3.06–25.3 |
| Extracorporeal membrane oxygenation | 7 (0.5) | 4 (<0.1) | 7 | 2.05–23.9 |
| Valvuloplasty | 2 (0.1) | 0 | - | - |
| Colectomy | 3 (0.2) | 2 (<0.1) | 6 | 1–35.9 |
| Gastrectomy | 3 (0.2) | - | - | - |
| Cholecystectomy | 8 (0.6) | 3 (<0.1) | 10.7 | 2.83–40.2 |
| Endoscopic retrograde cholangiopancreatography | 1 (<0.1) | 0 | - | - |
| Femur surgery | 5 (0.4) | 4 (<0.1) | 5 | 1.34–18.6 |

knowledge, this is the first report of an association between acute cardiac dysfunction diseases and NAION. The association between cardiac dysfunction and CVA has been well-explored and documented [10]. Both embryonically and anatomically, the optic nerves are rather an extension of the brain, and similarly a central retinal artery occlusion is considered a CVA equivalent. Although NAION does not constitute an embolic event, the relation of retinal tissue to brain tissue is well established, and therefore an association between NAION and cardiac dysfunction is reasonable [11].

We also found that the patients in the NAION group were 6 times more likely to have been diagnosed with hypovolemic events compared to the control group. Specifically, we revealed a 9-fold increase in the occurrence of a bleeding event in the month preceding the diagnosis of NAION. Vision loss following significant gastrointestinal bleeding has been observed historically, dating back to the era of Hippocrates [12]. Moreover, we also found a 12-fold increase in the occurrence of hemodialysis in the month preceding the diagnosis of NAION. Our findings are in accordance with those of Chang et al., who reported a 3-fold increase in the risk of NAION in individuals with end-stage renal disease. The authors attributed this elevated risk to fluctuations in blood pressure and reduced oxygen-carrying capacity during dialysis, leading to hypoperfusion and subsequent ischemia in the anterior segment of the optic nerve head [13, 14]. Our data indicate an even higher risk, which might be due to differences in the populations in the two studies. Our current study focused on patients on hemodialysis as well as those with end-stage renal disease, who might have greater fluctuations in blood pressure and therefore, might be at greater risk. We hypothesize that a similar mechanism may be attributed to the increased risk for NAION following major surgery, which is characterized by an acute drop in blood pressure and hypoperfusion following surgical blood loss. We found that patients with NAION were 8.8 times more likely to have undergone major surgery, such as coronary artery bypass graft (CABG) in the month prior to the diagnosis of NAION. This also supports previous findings. Shapira et al. followed 602 patients after coronary artery bypass graft surgery and found a 1.3% risk for NAION post-surgery [15]. During follow-up of 21 to 33 months, they observed that the vision deficit either remained unchanged or improved slightly. Shen et al. also examined ischemic optic neuropathy (ION) following major surgeries. They did not specifically differentiate between the anterior and posterior forms but noted that posterior ischemic optic neuropathy (PION) is the prevailing pattern following surgery [16].

The group of diseases characterized by a decrease in SVR as a cause of hypoperfusion was also found to be strongly associated with the occurrence of NAION, with a 4.64 fold increase. We found that patients diagnosed with NAION were 6.67 times more likely to have had the diagnosis of cholecystitis in the month preceding the diagnosis. Furthermore, pneumonia was also found to be associated with NAION, and patients diagnosed with NAION were 4.5 times more likely to have been diagnosed with pneumonia in the month preceding the diagnosis. Major surgery and infection can lead to reduced SVR, and though not documented it is reasonable to assume that at least some of the patients in these groups did experience an episode reduced SVR to a certain extent. Our findings support those of Weger et al., who investigated 71 patients and reported a 2.5-fold increase in the prevalence of IgG antibodies for *Chlamydia pneumoniae* in NAION patients compared to the control group, which further demonstrates the possible relation of an infection to NAION, presumably through the hypoperfusion caused by reduced SVR [17]. We also found a correlation between sepsis and NAION; however, due to few patients within the one-month timeframe, this did not reach statistical significance. Sepsis has been documented in association with NAION in several case reports, occurring 14 days after septic shock [18] and 14 days after sepsis caused by *Candida albicans* [19].

CVA was also identified as having a significant correlation with NAION, with patients diagnosed with NAION being 16 times more likely to have been diagnosed with a CVA in the month preceding the diagnosis. While some schools of thought equate NAION with a form of CVA, our investigation addressed CVA as a distinct event preceding the onset of NAION, adhering to the imposed one-month limitation in our analysis. Although previous studies have established CVA as a cardiovascular risk factor [6, 8], it can also trigger hypoperfusion. CVA is associated with substantial hemodynamic alterations, characterized by marked fluctuations in blood pressure that can result in hypotension and hypoperfusion. These alterations can occur through various mechanisms, including cerebral autoregulation [20] and sedation during endovascular stroke interventions [21].

## Strengths and limitations

This study investigated several risk factors for NAION on an extensive scale, reporting the largest cohort of patients to date, and while some of this study's findings were reported before by researchers such as Hayreh et al [8], the scale of this cohort allows to reinforce the relationship of hypoperfusion states and NAION.

However, the retrospective nature of the research poses inherent challenges, as reliance on medical records for data collection introduces the potential for inaccuracies or omissions due to varying record quality, along with the possibility of incorrect temporal alignment between events and diagnosis dates. Additionally, identifying cases using ICD-9 codes in medical records may result in missed diagnoses or misclassification. Nevertheless, the inclusion of a large study population helps mitigate the impact of these inaccuracies, temporal discrepancies and missing data by reducing random variation and strengthening the overall findings.

Finally, our study focused only on patients with the diagnosis of NAION. While adding to the cohort patients with the diagnosis of PION might have added to the size of the cohort and generalizability of our study, we found that the diagnosis of PION was inconsistently reported in patients' records.

## Conclusions

In conclusion, the results presented here reveal a significant association between hypoperfusion states and the onset of NAION. Conditions influencing cardiac output, such as hypovolemia and cardiac dysfunction, exhibited a more pronounced association with NAION

compared to those related to a decrease in SVR. In light of these findings, we recommend increasing awareness of NAION in the above mentioned conditions. We advise physicians in critical care follow-up clinics to inquire about any alterations in vision when evaluating patients after these conditions and refer them for urgent ophthalmologic assessment, if necessary.

## Supporting information

**S1 Table. List of ICD-9 codes.** List of comorbidities used for analysis.
(DOCX)

**S2 Table. Patient characteristics.** Underlying comorbidities and demographic data of patients not included in the matching process.
(DOCX)

## Author Contributions

**Conceptualization:** Eyal Walter, Erez Tsumi.

**Data curation:** Tomer Kerman, Nir Amitai, Itai Hazan, Ran Abuhasira.

**Formal analysis:** Jasmin Gabbay, Tomer Kerman, Nir Amitai, Itai Hazan, Ran Abuhasira.

**Investigation:** Jasmin Gabbay, Tomer Kerman, Ohad Gabay.

**Methodology:** Ohad Gabay, Ran Abuhasira.

**Writing – original draft:** Jasmin Gabbay, Nir Amitai, Ohad Gabay, Itai Hazan, Ran Abuhasira.

**Writing – review & editing:** Jasmin Gabbay, Eyal Walter, Erez Tsumi.

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
