## [Decision Letter · Decision Letter 0]

16 May 2024

PONE-D-24-16039Are Hypoperfusion States Risk factors for Non-arteritic Anterior Ischemic Optic Neuropathy?PLOS ONE

Dear Dr. Tsumi,

Thank you for submitting your manuscript to PLOS ONE. After careful consideration, we feel that it has merit but does not fully meet PLOS ONE’s publication criteria as it currently stands. Therefore, we invite you to submit a revised version of the manuscript that addresses the points raised during the review process.

We look forward to receiving your revised manuscript.

Kind regards,

Oana Dumitrascu, M.D.

Academic Editor

PLOS ONE

Journal Requirements:

Reviewers' comments:

Reviewer's Responses to Questions

**Comments to the Author**

1. Is the manuscript technically sound, and do the data support the conclusions?

Reviewer #1: Yes

Reviewer #2: Yes

2. Has the statistical analysis been performed appropriately and rigorously? 

Reviewer #1: Yes

Reviewer #2: Yes

3. Have the authors made all data underlying the findings in their manuscript fully available?

Reviewer #1: Yes

Reviewer #2: Yes

4. Is the manuscript presented in an intelligible fashion and written in standard English?

Reviewer #1: Yes

Reviewer #2: Yes

5. Review Comments to the Author

Reviewer #1: The authors present a compelling and interesting thesis that addresses an interesting question commonly raised but without current definitive evidence for a conclusion. This article suggests that hypovolemia in its several forms delineated by their team could be an independent risk factor for NAION. While retrospective in nature and unable to make a conclusive claim, it is provocative and worth publishing. I put in detailed comments into a separate word document, attached.

Reviewer #2: This review highlights and emphasizes the importance of hypoperfusion as risk factor for NAION with a clinically significant association. I’m not sure what is the reason the author mentioned the word “independent risk factor”.

6. PLOS authors have the option to publish the peer review history of their article (what does this mean?). If published, this will include your full peer review and any attached files.

Reviewer #1: No

Reviewer #2: **Yes: **Nafiseh Hashemi

---

## [Author Response · Author response to Decision Letter 0]

3 Jun 2024

Reviewer #1: The authors present a compelling and interesting thesis that addresses an interesting question commonly raised but without current definitive evidence for a conclusion. This article suggests that hypovolemia in its several forms delineated by their team could be an independent risk factor for NAION. While retrospective in nature and unable to make a conclusive claim, it is provocative and worth publishing. I put in detailed comments into a separate word document, attached.

Response: all comments were addressed accordingly.

Reviewer #2: This review highlights and emphasizes the importance of hypoperfusion as risk factor for NAION with a clinically significant association. I’m not sure what is the reason the author mentioned the word “independent risk factor”.

Response: Cardiovascular co-morbidities are well established risk factor for NAION. By using the term "independent risk factor" we have chosen to highlight the fact that states of hypoperfusion are still highly correlated with NAION even after stratifying the groups for these co-morbidities.

---

## [Decision Letter · Decision Letter 1]

6 Aug 2024

PONE-D-24-16039R1Hypoperfusion States Could Increase the Risk of Non-arteritic Anterior Ischemic Optic NeuropathyPLOS ONE

Dear Dr. Tsumi,

Thank you for submitting your manuscript to PLOS ONE. After careful consideration, we feel that it has merit but does not fully meet PLOS ONE’s publication criteria as it currently stands. Therefore, we invite you to submit a revised version of the manuscript that addresses the points raised during the review process.

The strength of this study lies in large number of subjects. However the presented findings in the manuscript are not unreported in published literature. Authors are suggested to provide a rationale for their study and state what new information their manuscript adds to the existing knowledge about role of hypoperfusion in nonarteritic ischemic optic neuropathy.

We look forward to receiving your revised manuscript.

Kind regards,

Kumar Saurabh

Academic Editor

PLOS ONE

Additional Editor Comments:

The strength of this study lies in large number of subjects. However the presented findings in the manuscript are not unreported in published literature. Authors are suggested to provide a rationale for their study and state what new information their manuscript adds to the existing knowledge about role of hypovolemia in nonarteritic ischemic optic neuropathy.

Reviewers' comments:

Reviewer's Responses to Questions

**Comments to the Author**

1. If the authors have adequately addressed your comments raised in a previous round of review and you feel that this manuscript is now acceptable for publication, you may indicate that here to bypass the “Comments to the Author” section, enter your conflict of interest statement in the “Confidential to Editor” section, and submit your "Accept" recommendation.

Reviewer #1: All comments have been addressed

Reviewer #3: All comments have been addressed

2. Is the manuscript technically sound, and do the data support the conclusions?

Reviewer #1: Yes

Reviewer #3: No

3. Has the statistical analysis been performed appropriately and rigorously? 

Reviewer #1: Yes

Reviewer #3: Yes

4. Have the authors made all data underlying the findings in their manuscript fully available?

Reviewer #1: Yes

Reviewer #3: Yes

5. Is the manuscript presented in an intelligible fashion and written in standard English?

Reviewer #1: Yes

Reviewer #3: Yes

6. Review Comments to the Author

Reviewer #1: Agree with the revisions and only one suggestion: change running title from Short title: Risk factors for non-arteritic anterior ischemic optic neuropathy to Short title: Hypoperfusion States in non-arteritic anterior ischemic optic neuropathy

Reviewer #3: Thank you for submitting the results of a large cohort of NAION patients.There are a few glaring flaws in the claims made in the paper.

1.This is not the first paper to address the association of the NAION to ischemic heart diseases.Authors have referenced Dr Hayrehs paper on the systemic diseases and the NAION(REF NO.8) that states" Also, middle-aged and elderly patients showed a significantly higher prevalence of ischemic heart disease (P < .01)"

The authors have chosen to selectively quote the articles details mentioning only the predisposing factors like HT,DM etc,at another place Dr Hayreh has clearly defined the predisposing factors and the precipitating factors in the evolution of the NAION.The hypoperfusion states of shock,be it the cardiogenic,hypovolemic etc are mentioned in the precipitating factors.

Hayreh, S S. Ischaemic optic neuropathy.. Indian Journal of Ophthalmology 48(3):p 171-194, Jul–Sep 2000.

The above paper clearly defines the following

Perfusion pressure = Mean BP minus intraocular pressure (IOP).

Mean BP = Diastolic BP + 1/3 (systolic minus diastolic BP).

Resistance to blood flow

This depends upon the state and calibre of the vessels supplying the ONH, which in turn are influenced by the following:

.....which includes systemic hypotension

Dr SS Hayreh has studied and extensively published about the pathophysiology of NAION.His work had already indicated what your study has observed in a large cohort.

I suggest that the introduction and the discussion be revised to present in a fair manner.If possible get the details of posterior and anterior ischemic optic neuropathy in the group,as the hypoperfusion associated with hypovolemia,is most likely to cause posterior ischemic optic neuropathy.

7. PLOS authors have the option to publish the peer review history of their article (what does this mean?). If published, this will include your full peer review and any attached files.

Reviewer #1: **Yes: **Kimberly Winges, MD

Reviewer #3: **Yes: **Shikha Talwar Bassi

---

## [Author Response · Author response to Decision Letter 1]

1 Sep 2024

Reviewer comments: 

Reviewer #1:

 Agree with the revisions and only one suggestion: change running title from Short title: Risk factors for non-arteritic anterior ischemic optic neuropathy to Short title: Hypoperfusion States in non-arteritic anterior ischemic optic neuropathy

Response:

Done, line 14

Reviewer #3:

Thank you for submitting the results of a large cohort of NAION patients.There are a few glaring flaws in the claims made in the paper.

1.This is not the first paper to address the association of the NAION to ischemic heart diseases.Authors have referenced Dr Hayrehs paper on the systemic diseases and the NAION(REF NO.8) that states" Also, middle-aged and elderly patients showed a significantly higher prevalence of ischemic heart disease (P < .01)"

The authors have chosen to selectively quote the articles details mentioning only the predisposing factors like HT,DM etc,at another place Dr Hayreh has clearly defined the predisposing factors and the precipitating factors in the evolution of the NAION.The hypoperfusion states of shock,be it the cardiogenic,hypovolemic etc are mentioned in the precipitating factors.

Hayreh, S S. Ischaemic optic neuropathy.. Indian Journal of Ophthalmology 48(3):p 171-194, Jul–Sep 2000.

The above paper clearly defines the following

Perfusion pressure = Mean BP minus intraocular pressure (IOP).

Mean BP = Diastolic BP + 1/3 (systolic minus diastolic BP).

Resistance to blood flow

This depends upon the state and calibre of the vessels supplying the ONH, which in turn are influenced by the following:

.....which includes systemic hypotension

Dr SS Hayreh has studied and extensively published about the pathophysiology of NAION.His work had already indicated what your study has observed in a large cohort.

I suggest that the introduction and the discussion be revised to present in a fair manner.If possible get the details of posterior and anterior ischemic optic neuropathy in the group,as the hypoperfusion associated with hypovolemia,is most likely to cause posterior ischemic optic neuropathy.

Response: 

We acknowledge that (to paraphrase off Isacc Newton) we stand on the shoulders of giants such as Dr SS Hayreh and that some of our findings were previously reported. The strength of our findings is in the large size of cohort, which can both help reach new conclusions and establish the validity of existing knowledge.

We have rephrased the relevant paragraph (lines 65-69) as well as rephrased the following sentence (lines 71-72).

Additionally- a comment was added in the discussion (lines 239-241).

With regards to diagnosing PION- we have decided to omit patients who were diagnosed with PION because we found that reporting of that diagnosis was inconsistent in patient’s files and did not allow a reliable large scale study. We added a comment in discussion (lines 247-250)

---

## [Decision Letter · Decision Letter 2]

18 Oct 2024

Hypoperfusion States Could Increase the Risk of Non-arteritic Anterior Ischemic Optic Neuropathy

PONE-D-24-16039R2

Dear Dr. Tsumi,

We’re pleased to inform you that your manuscript has been judged scientifically suitable for publication and will be formally accepted for publication once it meets all outstanding technical requirements.

Kind regards,

Kumar Saurabh

Academic Editor

PLOS ONE

Additional Editor Comments (optional):

Reviewers' comments:

Reviewer's Responses to Questions

**Comments to the Author**

1. If the authors have adequately addressed your comments raised in a previous round of review and you feel that this manuscript is now acceptable for publication, you may indicate that here to bypass the “Comments to the Author” section, enter your conflict of interest statement in the “Confidential to Editor” section, and submit your "Accept" recommendation.

Reviewer #1: All comments have been addressed

2. Is the manuscript technically sound, and do the data support the conclusions?

Reviewer #1: Yes

3. Has the statistical analysis been performed appropriately and rigorously? 

Reviewer #1: Yes

4. Have the authors made all data underlying the findings in their manuscript fully available?

Reviewer #1: Yes

5. Is the manuscript presented in an intelligible fashion and written in standard English?

Reviewer #1: Yes

6. Review Comments to the Author

Reviewer #1: (No Response)

7. PLOS authors have the option to publish the peer review history of their article (what does this mean?). If published, this will include your full peer review and any attached files.

Reviewer #1: No

---

## [Editor Report · Acceptance letter]

26 Oct 2024

PONE-D-24-16039R2 

PLOS ONE

Dear Dr. Tsumi, 

I'm pleased to inform you that your manuscript has been deemed suitable for publication in PLOS ONE. Congratulations! Your manuscript is now being handed over to our production team.

Kind regards, 

on behalf of

Dr. Kumar Saurabh 

Academic Editor

PLOS ONE